# LiDAR-PTQ: Post-Training Quantization for Point Cloud 3D Object Detection

**Sifan Zhou[1]* , Liang Li[2], Xinyu Zhang[2], Bo Zhang[2], Shipeng Bai[3]*, Miao Sun[4]**
**Ziyu Zhao[1], Xiaobo Lu[1]†, Xiangxiang Chu[2]†‡**
[1]Southeast University [2]Meituan Inc [3]Zhejiang University [4]Nanyang Technological University
sifanjay@gmail.com, xblu2013@126.com, chuxiangxiang@meituan.com

## Abstract

Due to highly constrained computing power and memory, deploying 3D lidar-based detectors on edge devices equipped in autonomous vehicles and robots poses a crucial challenge. Being a convenient and straightforward model compression approach, Post-Training Quantization (PTQ) has been widely adopted in 2D vision tasks. However, applying it directly to 3D lidar-based tasks inevitably leads to performance degradation. As a remedy, we propose an effective PTQ method called LiDAR-PTQ, which is particularly curated for 3D lidar detection (both SPConv-based and SPConv-free). Our LiDAR-PTQ features three main components, **(1)** a sparsity-based calibration method to determine the initialization of quantization parameters, **(2)** a Task-guided Global Positive Loss (TGPL) to reduce the disparity between the final predictions before and after quantization, **(3)** an adaptive rounding-to-nearest operation to minimize the layerwise reconstruction error. Extensive experiments demonstrate that our LiDAR-PTQ can achieve state-of-the-art quantization performance when applied to CenterPoint (both Pillar-based and Voxel-based). To our knowledge, for the very first time in lidar-based 3D detection tasks, the PTQ INT8 model's accuracy is almost the same as the FP32 model while enjoying $3\times$ inference speedup. Moreover, our LiDAR-PTQ is cost-effective being $30\times$ faster than the quantization-aware training method. Code will be released at https://github.com/StiphyJay/LiDAR-PTQ.

## 1 Introduction

LiDAR-based 3D detection has a wide range of applications in self-driving and robotics. It is important to detect the objects in the surrounding environment fastly and accurately, which places a high demand for both performance and latency. Currently, mainstream grid-based 3D detectors convert the irregular point cloud into arranged grids (voxels/pillars), and achieve top-ranking performance (Jiageng Mao, 2023) while facing a crucial challenge when deploying 3D lidar-based models on resource-limited edge devices. Therefore, it is important to improve the efficiency of grid-based 3D perception methods (e.g., reduce memory and computation cost).

Quantization is an efficient model compression approach for high-efficiency computation by reducing the number of bits for activation and weight representation. Compared to quantization-aware training (QAT) methods, which require access to all labeled training data and substantial computation resources, Post-training quantization (PTQ) is more suitable for fast and effective industrial applications. This is because PTQ only needs a small number of unlabeled samples as calibration set. Besides, PTQ does not necessitate retraining the network with all available labeled data, resulting in a shorter quantization process. Although several advanced PTQ methods (Nagel et al., 2020; Li et al., 2021; Wei et al., 2022; Yao et al., 2022) have been proposed for RGB-based detection tasks, applying it directly to 3D lidar-based tasks inevitably leads to performance degradation due to the differences between images and point clouds.

As shown in Fig 1, the inherent sparsity and irregular distribution of LiDAR point clouds present new challenges for the quantization of 3D Lidar-based detectors. **(1)** The sparsity of point cloud.

---

*Work done as an intern at Meituan. † Corresponding author. ‡Project leader.

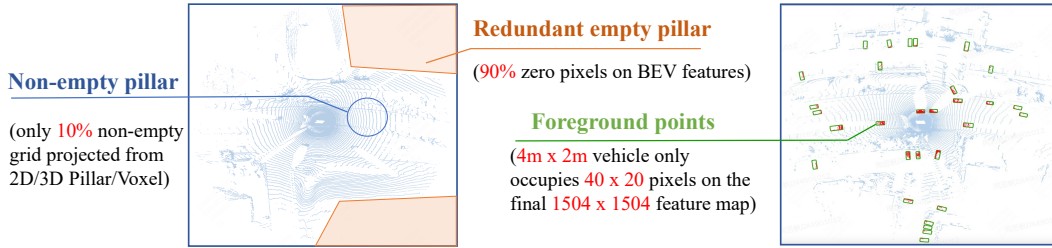

Figure 1: The sparsity of point cloud on 3D LiDAR-based object detection. Orange area means empty area, blue point means the point cloud (non-empty area) in a scenario, green box means the 3D Bboxes, and red point means foreground points.

Different from dense RGB images, non-zero pixels only occupy a very limited part of the whole scenario (about 10% in Waymo dataset (Sun et al., 2020)). For example, the huge number of zero pixel lead to significant differences in activation distribution compared to dense RGB-based tasks. **(2)** Larger arithmetic range. Compared with the 8-bit (0-255) RGB images, the point coordinates after voxelization are located in a $1504 \times 1504 \times 40$ (voxel size = 0.1m) 3D space in Waymo dataset, which makes it more susceptible to the effects of quantization (such as clipping error). **(3)** Imbalance between foreground instances and large redundant background area. For example, based on CenterPoint-Voxel (Yin et al., 2021), a vehicle with $4m \times 2m$ occupies only $40 \times 20$ pixels in the input $1504 \times 1504$ BEV feature map. Such small foreground instances and large perception ranges in 3D detection require the quantized model to have less information loss to maintain detection performance. Therefore, these challenges hinder the direct application of quantization methods developed for 2D vision tasks to 3D point cloud tasks.

To tackle the above challenge, we propose an effective PTQ method called LiDAR-PTQ, which is specifically curated for 3D LiDAR-based object detection tasks. Firstly, we introduce a sparsity-based calibration method to determine the initialization of quantization parameters on parameter space. Secondly, we propose Task-guided Global Positive Loss (TGPL) to find the quantization parameter on model space that is suitable for final output performance. Thirdly, we utilize an adaptive rounding value to mitigate the performance gap between the quantized and the full precision model. The proposed LiDAR-PTQ framework is a general and effective quantization method for both SPConv-based and SPConv-free 3D detection models. Extensive experiments on various datasets evaluate that our LiDAR-PTQ can achieve state-of-the-art quantization performance (Fig 2) when applied to CenterPoint (both Pillar-based and Voxel-based). To our knowledge, for the very first time in LiDAR-based 3D detection tasks, the PTQ INT8 model's accuracy is almost the same as the FP32 model while enjoying $3\times$ inference speedup. Moreover, our LiDAR-PTQ is cost-effective being $30\times$ faster than QAT method. We will release our code to the community.

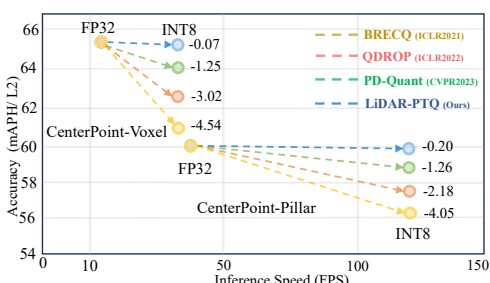

Figure 2: Performance comparison

Here, we summarize our main contributions as follows:

- Unveiling the root cause of performance collapse in the quantization of the 3D LiDAR-based detection model. Furthermore, we propose the sparsity-based calibration method to initialize the quantization parameter.

- TGPL: A Task-guided Global Positive Loss (TGPL) function to minimize the output disparity on model space which helps improve the quantized performance.

- LiDAR-PTQ: a general and effective quantization method for both SPConv-based and SPConv-free 3D detection models. Extensive experiments demonstrate LiDAR-PTQ can achieve state-of-the-art quantization performance on CenterPoint (both Pillar-based and Voxel-based).

- To our knowledge, for the very first time in LiDAR-based 3D detection tasks, the PTQ INT8 model's accuracy is almost the same as the FP32 model while enjoying $3\times$ inference speedup. Moreover, LiDAR-PTQ is cost-effective being $30\times$ faster than QAT method.

## 2 PRELIMINARIES

**LiDAR-based 3D object detection.** Given a point set with $N$ points in the 3D space, which is defined as $\mathbf{P} = \{\mathbf{p}_i = [x_i, y_i, z_i, r_i]^T \in \mathbb{R}^{N \times 4}\}$, where $x_i, y_i, z_i$ denote the coordinate values of each point along the axes X, Y, Z, respectively, and $r_i$ is the laser reflection intensity. Given a set of object in the 3D scene $\mathbf{B} = \{\mathbf{b}_j = [x_j, y_j, z_j, h_j, w_j, l_j, \theta_j, c_j]^T \in \mathbb{R}^{M \times 8}\}$, where $M$ is the total number of objects, $b_i$ is the $i$-th object in the scene, $x_j, y_j, z_j$ is the object's center, $h_j, w_j, l_j$ is the object's size, $\theta_j$ is the object's heading angle and $c_j$ is the object's class. The task of LiDAR-based 3D object detection is to detect the 3D boxes $\mathbf{B}$ from the point cloud $\mathbf{P}$ accurately.

**Quantization for tensor.** The quantization operation is defined as the mapping of a floating-point (FP) value $x$ (weights or activations) to an integer value $x_{int}$ according to the following equation:

$$x_{int} = clamp(\lfloor \frac{x}{s} \rceil + z, q_{min}, q_{max}) \tag{1}$$

where $\lfloor \cdot \rceil$ is the rounding-to-nearest operator, which results in the rounding error $\Delta r$. The function $clamp(\cdot)$ clips the values that lie outside of the integer range $[q_{min}, q_{max}]$, incurring a clipping error $\Delta c$. $x_{int}$ represents the quantized integer value. $z$ is zero-point. $s$ denotes the quantization scale factor, which reflects the proportional relationship between FP values and integers. $[q_{min}, q_{max}]$ is the quantization range determined by the bit-width $b$. Here, we adopt uniform signed symmetric quantization, as it is the most widely used in TensorRT (Migacz, 2017) and brings significant acceleration effect. Therefore, $q_{min} = -2^{b-1}$ and $q_{max} = 2^{b-1} - 1$. Nonuniform quantization (Jeon et al., 2022) is challenging to deploy on hardware, so we disregard it in this work. Generally, weights can be quantized without any need for calibration data. Therefore, the quantization of weights is commonly solved using grid search or analytical approximations with closed-form solution (Banner et al., 2019; Nagel et al., 2021) to minimize the mean squared error (MSE) in PTQ. However, activation quantization is input-dependent, so often requires a few batches of calibration data for the estimation of the dynamic ranges to converge. To approximate the real-valued input $x$, we perform the de-quantization step:

$$\hat{x} = (x_{int} - z) \cdot s \tag{2}$$

where $\hat{x}$ is the de-quantized FP value with an error that is introduced during the quantization process.

**Quantization range.** If we want to reduce clipping error $\Delta c$, we can increase the quantization scale factor $s$ to expand the quantization range. However, increasing $s$ leads to increased rounding error $\Delta r$ because $\Delta r$ lies in the range $\left[-\frac{s}{2}, \frac{s}{2}\right]$. Therefore, the key problem is how to choose the quantization range $(x_{min}, x_{max})$ to achieve the right trade-off between clipping and rounding error. Specifically, when we set fixed bit-width $b$, the quantization scale factor $s$ is determined by the quantization range:

$$s = (x_{max} - x_{min}) / (2^b - 1) \tag{3}$$

There are two common methods for quantization range setting.

i): *Max-min calibration.* We can define the quantization range as:

$$x_{max} = max(|x|), x_{min} = -x_{max} \tag{4}$$

to cover the whole dynamic range of the floating-point value $x$. This leads to no clipping error. However, this approach is sensitive to outliers as strong outliers may cause excessive rounding errors.

ii): *Entropy calibration.* TensorRT (Migacz, 2017) minimize the information loss between $x$ and $\hat{x}$ based on the KL divergence to determine the quantization range:

$$\underset{x_{min}, x_{max}}{\arg\min} \ D_{KL}(x, \hat{x}) \tag{5}$$

where $D_{KL}$ denotes the Kullback-Leibler divergence function. The entropy calibration will saturate the activations above a certain threshold to remove outliers. More details refer to the appendix.

**Quantization for network.** For a float model with $N$ layer, we primarily focus on the quantization of convolutional layers or linear layers, which mainly involves the handling of weights and activations.

For a given layer $L_i$, we initially execute quantization operations on its weight and input tensor, as illustrated in Eq 14 and 2, yielding $\hat{W}_i$ and $\hat{I}_i$. Consequently, the quantized output of this layer can be expressed as follows.

$$\hat{A}_i = f(BN(\hat{I}_i \circledast \hat{W}_i)) \tag{6}$$

where $\circledast$ denotes the convolution operator, $BN(\cdot)$ is the Batch-Normalization procedure, and $f(\cdot)$ is the activation function. Quantization works generally take into account the convolution, Batch Normalization (BN), and activation layers.

## 3 METHODOLOGY

Here, we first conduct PTQ ablation study on the CenterPoint-Pillar (Yin et al., 2021) model using two different calibrators (Entropy and Max-min) on Waymo $val$ set. As shown in Table 1, when using INT8 quantization, the performance drop is severely compromised for both the calibration method, especially for the entropy calibrator with a significant accuracy drop of **-38.67 mAPH/L2**.

However, directly employing the Max-min calibrator yielded better results, yet not unsatisfactory. It is entirely contrary to our experience in 2D model quantization, where entropy calibration effectively mitigates the impact of outliers, thereby achieving superior results (Nagel et al., 2021). Similar observations are also discussed in Stäcker et al. (2021). This anomaly prompts us to propose a general and effective PTQ method for 3D LiDAR-based detectors. In Waymo dataset, the official evaluation tools evaluated the methods in two difficulty levels: LEVEL_1 for boxes with more than five LiDAR points, and LEVEL_2 for boxes with at least one LiDAR point. Here we report the metrics in Mean Average Precision with Heading / LEVEL_2 (mAPH/L2), which is a widely adopted metric by the community.

Table 1: Ablation study.

| Method | Bits(W/A) | LEVEL_2 mAPH | | |
| --- | --- | --- | --- | --- |
| | | Mean | Vehicle | Pedestrian |
| Full Prec. | 32/32 | 60.32 | 65.42 | 55.23 |
| Entropy | 8/8 | 21.65 **(-38.67)** | 29.41 **(-36.02)** | 11.89 **(-43.34)** |
| Max-Min | 8/8 | 52.91 **(-7.41)** | 55.37 **(-10.05)** | 50.45 **(-4.78)** |

### 3.1 LiDAR-PTQ FRAMEWORK

In this paper, we propose a post-training quantization framework for point cloud models, termed LiDAR-PTQ. Our LiDAR-PTQ could enable the quantized model to achieve almost the same performance as the FP mode, and there is no extra huge computation cost and access to labeled training data. This framework primarily comprises three components.

**i) Sparsity-based calibration**: We employ a Max-min calibrator equipped with a lightweight grid search to appropriately initialize the quantization parameters for both weights and activations.

**ii) Task-guided Global Positive Loss (TGPL)**: This component utilizes a specially designed foreground-aware global supervision to further optimize the quantization parameters of activation.

**iii) Adaptive rounding-to-nearest**: This module aims to mitigate the weight rounding error $\Delta r$ by minimizing the layer-wise reconstruction error.

In summary, our method first initializes the quantization parameters for weights and activations through a search in the parameter space, and then further refine them through a process of supervised optimization in the model space. Consequently, our method is capable of achieving a quantized accuracy that almost matches their float counterparts for certain lidar detectors.

We formulate the our LiDAR-PTQ algorithm for a full precision 3D detector in Algorithm 2. Next, we will provide detailed explanations for these three parts.

### 3.2 SPARSITY-BASED CALIBRATION

Here, in order to delve into the underlying reasons for the huge performance gap (31.29 mAPH/L2 in Tab 1) between Max-min and entropy calibrator. We statistically analyze the numerical distribution of feature maps of both RGB-based models and LiDAR-based object detection models, and visualize the main diversity as shown in Fig 3. The main reasons affecting the quantization performance can be summarized in two points:

---

**Algorithm 1** LiDAR-PTQ quantization

---

**Input**: Pretrained FP model with $N$ layers; Calibration dataset $D^c$, iteration $T$.
**Output**: quantization parameters of both activation and weight in network, i.e., weight scale $s_w$, weight zero-point $z_w$, activation scale $s_a$, activation zero-point $z_a$ and adaptive rounding value for weight $\theta$.

1: Optimize only weight quantization parameters $s_w$ and $z_w$ to minimize Eq 16 in every layer using the grid search algorithm;
2: input $D^c$ to FP network to get the FP final output $O_{fp}$
3: **for** $L_n = \{L_i | i = 1, 2, ...N\}$ **do**
4:     Optimize only activation quantization parameters $s_a$ and $z_a$ to minimize Eq 16 in layer $L_i$ using the grid search algorithm;
5:     Collect input data $I_i$ to the FP layer $L_i$;
6:     Input $I_i$ to quantized layer $L_i^q$ and FP layer $L_i$ to get quantized output $\hat{A}_i$ and FP output $A_i$;
7:     Input $\hat{A}_i$ to the following FP network to get output $\hat{O_{par}}$ of partial-quantized network;
8:     **for** all $j = 1, 2, \ldots, T$-iteration **do**
9:         Check quantized output $\hat{A}_i$ and FP output and calculate $L_{local}$ using Eq 11;
10:         Check partial-quantized network output $\hat{O_{par}}$ and FP final output $O_{fp}$ to calculate $L_{TGPL}$ using Eq 9;
11:         Optimize quantization parameters $s_w, z_w, s_a$, and $z_a, \theta$ of layer $L_i$ to minimize $L_{total}$ using Eq 12;
12:     **end for**
13:     Quantize layer $L_i$ with the learnable quantization parameters $s_w, z_w, s_a$, and $z_a, \theta$;
14: **end for**

---

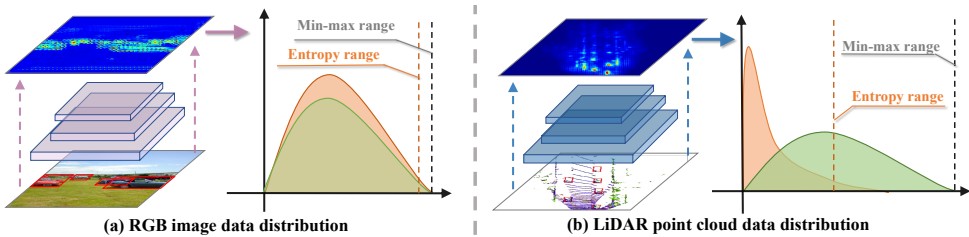

**(a) RGB image data distribution**              **(b) LiDAR point cloud data distribution**

Figure 3: The diagram of data distribution for RGB-based and LiDAR-based object detection. Orange and green denote the data distribution of the entire feature map and foreground feature.

**i) Huge sparsity lead to inappropriate quantization range.** As shown in Fig 1 and 3, the sparsity of point cloud makes the whole BEV feature map exist a large number of zero pixels. Therefore, the entropy calibrator will statistic the feature value including zero pixels ($\approx 90\%$) to minimize the information loss, which leads to the values outside the quantization range being clipped. However, these truncated values contain rich geometric representations that could used for final object detection.

**ii) Point cloud features are more sensitive to quantization range.** Point cloud explicitly gauges spatial distances, and shapes of objects by collecting laser measurement signals from environments. During voxelization process, the raw point cloud coordinates, i.e., $x, y, z$ in the ego-vehicle coordinate system are encoded as part of voxel features that preserve essential geometric information. Specifically, the arithmetic range of the input point cloud coordinates increases with detection distance. Therefore, the arithmetic range in the voxel feature is strongly correlated with detection distance. In other words, the arithmetic range of point cloud is relevant to the geometrics.

Furthermore, we also conduct an ablation study with different range distances on waymo *val* set. As shown in Tab 2, we find that *the decline in accuracy is exacerbated as the distance increases.*
For entropy calibrator, the quantized performance on long-range metrics (50m - inf) is terribly damaged (5.90 mAPH/L2, up to 84.5% drop), while accuracy on short-range

Table 2: Ablation study in different range .

| Method | Bits(W/A) | Vehicle LEVEL_2 mAPH | | |
| | | Mean 0-30 m | 30-50 m | 50-$\infty$ m |
|---|---|---|---|---|
| Full Prec. | 32/32 | 88.77 | 65.23 | 38.09 |
| Entropy | 8/8 | 60.03 (**-28.74**) (**-32.4%**) | 22.46 (**-42.77**) (**-65.6%**) | 5.90 (**-32.19**) (**-84.5%**) |
| Max-min | 8/8 | 87.14 (**-1.63**) (**-1.8%**) | 57.49 (**-7.74**) (**-11.9%**) | 22.98 (**-15.11**) (**-39.7%**) |

metrics (0-30m) remains well (60.03 mAPH/L2, 32.4% drop). This is because the entropy calibrator

provides an inappropriate quantization range, resulting in a significant clipping error. Therefore, a large number of values with geometrics info are truncated and consequently a substantial degradation in model accuracy. On the contrary, for the Max-min calibrator, which covers the whole dynamic range of the FP activation, the values with geometric information are preserved effectively. Therefore, its performance in different range metrics performs well, especially on short-range metrics (0-30m), which only drops 1.63 mAPH/L2 (1.8%) than FP model.

Drawing upon the above findings, we conclude that the commonly used calibration method for RGB images is sub-optimal, while Max-min is more suitable for 3D point clouds. Therefore, we adopt Max-min calibrator for both weights and activations to mitigate the impact of high sparsity. Besides, to get more finer-grained scale factor and avoid the influence of outliers on rounding error $\Delta r$, a lightweight grid search (Banner et al., 2019; Choukroun et al., 2019) is incorporated to further optimize the quantization parameters.

Specifically, for a weight or activation tensor $X$, firstly obtain the $x_{max}$ and $x_{min}$ according to Eq.4, and calculate the initial quantization parameter $s_0$ following Eq. 15. Then linearly divide the interval $[\alpha s_0, \beta s_0]$ into $T$ candidate bins, denoted as $\{s_t\}_{t=1}^T$. $\alpha$, $\beta$ and $T$ are designed to control the search range and granularity. Finally, search $\{s_t\}_{t=1}^T$ to find the optimal $s_{opt}$ that minimizes the quantization error,

$$\arg\min_{s_t} \|(X - \hat{X}(s_l))\|_F^2 \tag{7}$$

$\|\cdot\|_F^2$ is the Frobenius norm (MSE Loss). Refer to appendix for more details about grid search.

## 3.3 TASK-GUIDED GLOBAL POSITIVE LOSS

The aforementioned calibration initialization approach can effectively improve the quantization accuracy of lidar detectors, but there is still a significant gap compared with the float model.

Both empirical and theoretical evidence (Li et al., 2021; Wei et al., 2022; Liu et al., 2023a) suggest that solely minimizing the quantization error in parameter space dose not guarantee equivalent minimization in final task loss within model space. Therefore, it becomes imperative to devise a global supervisory signal specifically tailored for 3D LiDAR-based detection tasks. This supervision would enable further fine-tuning of the quantization parameters $p$ to achieve higher quantized precision. It is crucial to emphasize that this fine-tuning process does not involve labeled training data. Only need to minimize the distance between float output $O^f$ and the quantized model's output $\hat{O}$, as depicted in Eq 8

$$\arg\min_{p}(O^f - \hat{O}) \tag{8}$$

In this paper, we propose Task-guided Global Positive Loss (TGPL) function to constrain the output disparity between the quantized and FP models. Our TGPL function features two characteristics that contribute to improving the performance of the quantized method:

**i) Optimal quantization parameter on model space.** The TGPL function compares the final output difference between the FP and the quantized models rather than the difference in each layer's output.

**ii) Task-guided.** As mentioned in Sec1 and Fig 1, there exists extreme imbalance between small informative foreground instances and large redundant background areas in Lidar-based detection tasks. For sparse 3D scenes, it is sub-optimal to imitate all feature pixels on dense 2D images. TGPL function is designed to leverage cues in the FP model's classification response to guide the quantized model to focus on the important area (*i.e.* positive sample location) that is relevant to final tasks.

In detail, we filter all prediction boxes from the FP model by a threshold $\gamma$, then we select the top $K$ boxes. Then we perform NMS (Neubeck & Van Gool, 2006) to get the final prediction as positive boxes (pusedo-labels). Specifically, inspired by the Gaussian label assignment in CenterPoint (Yin et al., 2021), we define positive positions in a soft way with center-peak Gaussian distribution. Finally, for the classification branch, we use the focal loss (Lin et al., 2017) as the heatmap loss $\mathcal{L}_{cls}$. For the 3D box regression, we make use of the L1 loss $\mathcal{L}_{reg}$ to supervise their localization offsets, size and orientation. The overall TGPL loss consists of two parts as follows:

$$\mathcal{L}_{TGPL} = \mathcal{L}_{cls} + \alpha\mathcal{L}_{reg}, \tag{9}$$

## 3.4 ADAPTIVE ROUNDING-TO-NEAREST

Through grid search initialization and TGPL function constrain, the performance of quantized model has been greatly improved, but there is still a gap in achieving comparable accuracy with FP model. Recently, some methods (Wei et al., 2022; Liu et al., 2023a) optimize a variable, called rounding values, to determine whether weight values will be rounded up or down during the quantization process. In this way, the Eq 14 in weight quantization can be formulated as follows:

$$x_{int} = clamp(\lfloor \frac{x + \theta}{s} \rceil + z, q_{min}, q_{max}),$$ (10)

where $\theta$ is the optimization variable for each weight value to decide rounding results up or down (Nagel et al., 2020), *i.e.*, $\frac{\theta}{S}$ ranges from 0 to 1. Inspired by AdaRound(Nagel et al., 2020), we add a local reconstruction item to help learn the rounding value $\theta$. The local reconstruction item as follows:

$$L_{Local} = \|(W_i \circledast I_i - \hat{W}_i \circledast I_i)\|_F^2$$ (11)

where $\| \cdot \|_F^2$ is the Frobenius norm and $\hat{W}_i$ are the soft-quantized weights are calculated by Eq 10 and Eq 2. This operation allows us to adapt the rounding value to minimize information loss according to the calibration data, ensuring that the quantization process preserves important details. By adjusting the rounding value, we can achieve better performance of LiDAR-PTQ. Finally, the overall loss of our LiDAR-PTQ consists of two parts as follows:

$$\mathcal{L}_{total} = \lambda_1 \mathcal{L}_{local} + \lambda_2 \mathcal{L}_{TGPL},$$ (12)

## 4 EXPERIMENTS

**Dataset.** To evaluate the effectiveness of our proposed Lidar-PTQ, we conduct main experiments on large-scale autonomous driving datasets, Waymo Open Dataset (WOD) (Sun et al., 2020).

**Implementation Details.** In WOD dataset, we randomly sample 256 frames point cloud data from the training set as the calibration data. The calibration set proportions is **0.16%** (256/158,081) for WOD. We set the first and the last layer of the network to keep full precision. The learning rate for the activation quantization scaling factor is 5e-5, and for weight quantization rounding is 5e-3. In TGPL loss, we set $\gamma$ as 0.1, and K as 500. More details in supplements.

### 4.1 PERFORMANCE COMPARISON ON WAYMO DATASET

Table 3: Performance comparison on Waymo *val* set. ‡: reimplementation by us.

| Models | Methods | Bits(W/A) | Mean (L2) | | Vehicle (L2) | | Pedestrian (L2) | | Cyclist (L2) | |
|---|---|---|---|---|---|---|---|---|---|---|
| | | | mAP | mAPH | mAP | mAPH | mAP | mAPH | mAP | mAPH |
| | Full Prec. | 32/32 | 65.78 | 60.32 | 65.92 | 65.42 | 65.65 | 55.23 | - | - |
| CP-Pillar | BRECQ | 8/8 | 61.73 | 56.27 | 61.87 | 61.36 | 61.59 | 51.18 | - | - |
| | QDROP | 8/8 | 63.60 | 58.14 | 63.74 | 63.23 | 63.46 | 53.04 | - | - |
| | PD-QUANT | 8/8 | 64.59 | 59.06 | 64.87 | 64.21 | 64.32 | 53.91 | - | - |
| | QAT‡ | 8/8 | 65.56 | 60.08 | 65.69 | 65.17 | 65.44 | 54.99 | - | - |
| | **LiDAR-PTQ** | 8/8 | **65.60** | **60.12** | **65.64** | **65.14** | **65.55** | **55.11** | - | - |
| | Full Prec. | 32/32 | 67.67 | 65.25 | 66.29 | 65.79 | 68.04 | 62.35 | 68.69 | 67.61 |
| CP-Voxel | BRECQ | 8/8 | 63.15 | 60.71 | 62.53 | 62.03 | 63.22 | 57.49 | 63.71 | 62.60 |
| | QDROP | 8/8 | 64.70 | 62.23 | 63.97 | 63.38 | 64.90 | 59.17 | 65.24 | 64.13 |
| | PD-QUANT | 8/8 | 66.45 | 64.00 | 65.11 | 64.56 | 66.91 | 61.18 | 67.32 | 66.25 |
| | QAT‡ | 8/8 | 67.63 | 65.20 | 66.28 | 65.76 | 67.98 | 62.28 | 68.62 | 67.55 |
| | **LiDAR-PTQ** | 8/8 | **67.60** | **65.18** | **66.27** | **65.78** | **67.95** | **62.24** | **68.60** | **67.52** |

Due to there are no PTQ methods specially designed for 3D LiDAR-based detection tasks, we reimplement several advanced PTQ methods in 2D RGB-based vision tasks, which are BRECQ (Li et al., 2021), QDROP (Wei et al., 2022) and PD-Quant (Liu et al., 2023a). Specifically, we select well-known CenterPoint (Yin et al., 2021) as our full precision model and report the quantized performance on WOD (Sun et al., 2020) dataset. Because it includes SPConv-based and SPConv-free models, which could effectively verify the generalization of our LiDAR-PTQ. As shown in Tab 3, LiDAR-PTQ achieves state-of-the-art performance and outperforms BRECQ and QDrop by a large margin of 3.87 and 2.00 on CenterPoint-Pillar model and 4.45 and 2.90 on CenterPoint-Voxel model.

For PD-Quant, a state-of-the-art PTQ method specially designed for RGB-based vision tasks, but it has suboptimal performance on LiDAR-based tasks. Specifically, to solve the over-fitting problem on the calibration set, PD-Quant adjusts activation according to FP model's BN layer. However, for the point cloud which is more sensitive to the arithmetic range, this design is ineffective and time-consuming, and will lead to accuracy loss. Notably, our LiDAR-PTQ achieves on-par or even superior accuracy than the QAT model and almost without performance drop than the float model.

## 4.2 THE EFFECTIVENESS OF LiDAR-PTQ FOR FULLY SPARSE DETECTOR

Table 4: The performance of FSD on Waymo *val* set.

| Models | Methods | Bits(W/A) | Mean (L2) | | Vehicle (L2) | | Pedestrian (L2) | | Cyclist (L2) | |
|---|---|---|---|---|---|---|---|---|---|---|
| | | | mAP | mAPH | mAP | mAPH | mAP | mAPH | mAP | mAPH |
| | Full Prec. | 32/32 | 73.01 | 70.94 | 70.34 | 69.98 | 73.95 | 69.16 | 74.75 | 73.69 |
| FSD | Entropy | 8/8 | 10.54 | 9.44 | 0.06 | 0.06 | 21.88 | 18.86 | 9.69 | 9.41 |
| | Max-min | 8/8 | 71.24 | 69.37 | 68.42 | 68.18 | 72.09 | 67.60 | 73.22 | 72.34 |
| | **LiDAR-PTQ** | 8/8 | **72.84** | **70.73** | **69.95** | **69.62** | **73.85** | **68.95** | **74.71** | **73.63** |

Recently, there are emerging of some fully sparse 3D detectors, like FSD (Fan et al., 2022), FSD++ (Fan et al., 2023) and VoxelNext (Chen et al., 2023), etc. Here, we take FSD as an example to validate the effectiveness of our LiDAR-PTQ on fully sparse detectors. As shown in Tab 4, adopting entropy calibration still leads to a significant accuracy drop of **-61.50**. We discover that quantized FSD readily delivers the desired performance while employing a vanilla max-min calibration. Nonetheless, using LiDAR-PTQ can further achieve comparable accuracy to its float counterpart. The experiments demonstrate that LiDAR-PTQ is also applicable to fully sparse detectors.

## 4.3 ABLATION STUDY

Table 5: Ablation study of different components of LiDAR-PTQ on Waymo *val* set.

| Models | Methods | Bits(W/A) | Mean (L2) | | Vehicle (L2) | | Pedestrian (L2) | |
|---|---|---|---|---|---|---|---|---|
| | | | mAP | mAPH | mAP | mAPH | mAP | mAPH |
| | Full Prec. | 32/32 | 65.78 | 60.32 | 65.92 | 65.42 | 65.65 | 55.23 |
| CP-Pillar | Max-min | 8/8 | 57.33 | 52.91 | 55.64 | 55.37 | 59.02 | 50.45 |
| | +GRID S | 8/8 | 63.66 | 58.39 | 63.37 | 62.87 | 63.96 | 53.91 |
| | +TGPL | 8/8 | 64.81 | 59.40 | 65.12 | 64.53 | 64.50 | 54.27 |
| | +Round | 8/8 | **65.60** | **60.12** | **65.64** | **65.14** | **65.55** | **55.11** |

Here, we conduct ablation study of different components in our LiDAR-PTQ based on CenterPoint-Pillar model to verify their effects. As shown in Tab 5, based on the selected Max-min calibrator, we could obtain 5.48 mAPH/L2 performance gain by using a lightweight grid search method. However, grid search only minimizes reconstruction error in parameter space, which is not equivalent to minimize the final performance loss. Therefore, by introducing the proposed TGPL function to fine-tune quantization parameters in model space, the performance of quantized model coud be 59.40 mAPH/L2. Finally, by introducing an adaptive rounding value, a freedom degree (Eq 10) is added to mitigate the final performance gap and achieve almost the same performance as the FP model (60.12 vs 60.32). Notably, the performance of FP model is the upper limit of the quantized model because we focus on post-training quantization without labeled training data.

## 4.4 INFERENCE ACCELERATION

Here, we compared the speed of CenterPoint before and after quantization on an NVIDIA Jeston AGX Orin. This is a resource-constrained edge GPU platform that is widely used in real-word self-driving cars. The speed of quantized model enjoying $3\times$ inference speedup, which demonstrates that our LiDAR-PTQ can effectively improve the efficiency of 3D detection model on edge devices.

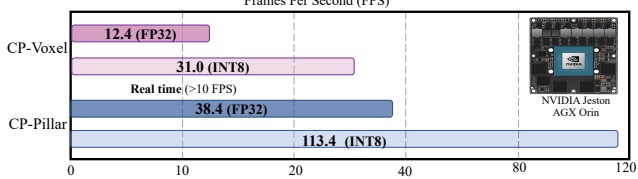

| Method | Bits(W/A) | Latency | FPS |
|---|---|---|---|
| CP-Pillar | 32/32 | 26.01 | 38.4 |
| | 8/8 | 8.82 | 113.4 |
| CP-Voxel | 32/32 | 80.64 | 12.4 |
| | 8/8 | 32.26 | 31.0 |

## 4.5 COMPUTATION EFFICIENCY

LiDAR-PTQ requires additional computation and fine-tuning process compared to other traditional PTQ methods, resulting in increased time costs. While the quantization time is a limitation of LiDAR-PTQ, compared with other advanced PTQ methods, LiDAR-PTQ's additional time cost is acceptable.

Furthermore, QAT method, the quantization time of LiDAR-PTQ is very short. For example, CenterPoint-Pillar will take 94 GPU/hour to achieve the same performance

| Model | QAT | BRECQ | QDROP | PD-QUANT | LiDAR-PTQ |
|---|---|---|---|---|---|
| CP-Pillar | 93.82 | 1.93 | 1.82 | 6.72 | 2.75 |
| CP-Voxel | 80.51 | 1.73 | 1.64 | 6.13 | 2.12 |

as the FP model on WOD dataset, while LiDAR-PTQ takes only 3 GPU/hour, which is $30 \times$ faster than the QAT method. It also proves that our LiDAR-PTQ is cost-effective.

## 5 RELATED WORKS

**Post-training Quantization (PTQ).** As mentioned in (Krishnamoorthi, 2018), existing quantization methods can be divided into two categories: (1) Quantization-Aware Training (QAT) and (2) Post-Training Quantization (PTQ). QAT methods (Wei et al., 2018; Li et al., 2019; Esser et al., 2019; Zhuang et al., 2020; Chen et al., 2021) require access to all labeled training data, which may not be feasible due to data privacy and security concerns. Compared to Quantization-aware Training (QAT) methods, Post-training quantization (PTQ) methods are simpler to use and allow for quantization with limited unlabeled data. Currently, there are many methods (Wu et al., 2020; Nahshan et al., 2021; Yuan et al., 2022; Liu et al., 2023b; Chu et al., 2024) designed for 2D vision tasks. AdaRound (Nagel et al., 2020) formulates the rounding task as a layer-wise quadratic unconstrained binary optimization problem and achieves a better performance. Based on AdaRound, BRECQ (Li et al., 2021) proposes utilizing block reconstruction to further enhance the accuracy of post-training quantization (PTQ). After that, QDrop (Wei et al., 2022) randomly drops the quantization of activations during PTQ and achieves new state-of-the-art accuracy. PD-Quant (Liu et al., 2023a) considers the global difference of models before and after quantization and adjusts the distribution of activation by BN layer statistics to alleviate the overfitting problem. However, these methods are specially designed for RGB images, and they are not readily transferable to LiDAR point cloud with substantial modal differences.

**Quantization for 3D Object Detection.** With the wide application of 3D object detection, in autonomous driving and robotics, some quantization methods are designed to improve inference speed for onboard deployment applications. With the advance of quantization techniques based on RGB image, QD-BEV (Zhang et al., 2023) achieves smaller size and faster speed than baseline BevFormer (Li et al., 2022) using QAT and distillation on multi-camera 3D detection tasks. For LiDAR-based 3D detection, especially for fully convolutional methods, like PointPillars (Lang et al., 2019), FCOS-LIDAR (Tian et al., 2022), FastPillars (Zhou et al., 2023), etc., effective quantization solutions could significantly speedup their latency to meet the practical requirements. (Stäcker et al., 2021) find that directly using INT8 quantization for 2D CNN will bring significant performance drop on PointPillars (Lang et al., 2019), where the reduction is even more severe for the entropy calibrator. Besides, BiPointNet (Qin et al., 2021) is a binarization quantization method, which focuses on classification and segmentation tasks based on point cloud captured from small CAD simulation. To the best of our knowledge, there is no quantization solution designed for large-scale outdoor LiDAR-based 3D object detection methods in self-driving.

## 6 CONCLUSION AND FUTURE WORK

In this paper, we analyze the root cause of the performance degradation of point cloud data during the quantization process. Then we propose an effective PTQ method called LiDAR-PTQ, which is particularly designed for 3D LiDAR-based object detection tasks. Our LiDAR-PTQ features three main components: **(1)** a sparsity-based calibration method to determine the initialization of quantization parameters **(2)** a Task-guided Global Positive Loss (TGPL) to reduce the disparity on the final task. **(3)**. an adaptive rounding-to-nearest operation to minimize the layer-wise reconstruction error. Extensive experiments demonstrate that our LiDAR-PTQ can achieve state-of-the-art performance on CenterPoint (both pillar-based and voxel-based). To our knowledge, for the very first time in lidar-based 3D detection tasks, the PTQ INT8 model's accuracy is almost the same as the FP32 model while enjoying $3\times$ inference speedup. Moreover, our LiDAR-PTQ is cost-effective being $30\times$ faster than the quantization-aware training method. Given its effectiveness and efficiency, we hope that our LiDAR-PTQ can serve as a valuable quantization tool for current mainstream grid-based 3D detectors and push the development of practical deployment of 3D detection models on edge devices. Besides, we believe that the low-bit quantization of 3D detectors will bring further efficiency improvement. This remains an open problem for future research.

ACKNOWLEDGMENTS

We thank anonymous reviewers for their kind help of this work. This work was supported by National Key R&D Program of China (No. 2022ZD0118700), National Natural Science Foundation of China (No.62271143), and the Big Data Computing Center of Southeast University.

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

## APPENDIX A: LiDAR-PTQ FOR DIFFERENT DETECTORS

CenterPoint (Yin et al., 2021) integrates two milestone works in LiDAR-based BEV detection, VoxelNet (Zhou & Tuzel, 2018) and PointPillars (Lang et al., 2019) as CP-Pillar and CP-Voxel. In particular, CP-Pillar and CP-Voxel have different network design. The CP-Pillar model is a fully dense convolutional network, while the CP-Voxel model includes SP-Conv and dense convolution. Our results on CenterPoint (-pillar and -voxel) demonstrate that: **i)** Lidar-PTQ is applicable to pillar-based and voxel-based detectors. **ii)** Lidar-PTQ is applicable to SPConv and dense convolution operations.

| Method | representation | backbone | neck | head |
|---|---|---|---|---|
| CP-Pillar | Pillar | dense | dense | dense |
| CP-Voxel | Voxel | sparse | dense | dense |
| FSD | Point+Voxel | sparse | sparse | sparse |

Table 6: Performance comparison on nuScene *val* set. We show the NDS, and mAP for each class. Abbreviations: construction vehicle (CV), pedestrian (Ped), motorcycle (Motor), bicycle (BC) and traffic cone (TC).

| Models | Methods | Bits(W/A) | NDS | mAP | Car | Truck | Bus | Trailer | CV | Ped | Motor | BC | TC | Barrier |
|---|---|---|---|---|---|---|---|---|---|---|---|---|---|---|
| | Full Prec. | 32/32 | 60.3 | 50.0 | 83.8 | 50.6 | 61.8 | 31.2 | 9.2 | 79.4 | 44.1 | 20.2 | 57.7 | 61.3 |
| CP-Pillar | BRECQ | 8/8 | 56.9 | 43.6 | 75.9 | 41.4 | 54.3 | 21.6 | 3.8 | 78.1 | 37.4 | 15.7 | 55.0 | 53.3 |
| | QDROP | 8/8 | 57.8 | 45.9 | 78.8 | 44.2 | 57.0 | 23.8 | 5.2 | 78.4 | 40.1 | 17.6 | 56.7 | 56.8 |
| | PD-QUANT | 8/8 | 59.6 | 48.3 | 81.8 | 47.6 | 59.4 | 28.2 | 7.8 | 78.4 | 41.6 | 19.8 | 57.6 | 61.0 |
| | **LiDAR-PTQ** | 8/8 | **60.2** | **49.8** | **83.7** | **50.8** | **61.8** | **30.6** | **9.0** | **79.0** | **43.6** | **20.4** | **57.8** | **61.0** |
| | Full Prec. | 32/32 | 64.8 | 56.6 | 84.6 | 54.5 | 66.7 | 36.4 | 16.9 | 83.1 | 56.1 | 39.6 | 64.0 | 64.3 |
| CP-Voxel | BRECQ | 8/8 | 62.0 | 51.2 | 76.5 | 46.8 | 60.5 | 28.9 | 12.5 | 80.4 | 53.7 | 34.8 | 58.8 | 59.1 |
| | QDROP | 8/8 | 63.2 | 54.0 | 82.1 | 48.5 | 64.9 | 32.9 | 15.1 | 81.1 | 55.1 | 36.9 | 60.9 | 63.7 |
| | PD-QUANT | 8/8 | 63.7 | 55.2 | 83.7 | 51.1 | 66.6 | 34.1 | 16.6 | 82.8 | 55.1 | 36.4 | 62.6 | 63.2 |
| | **LiDAR-PTQ** | 8/8 | **64.7** | **56.5** | **84.6** | **54.2** | **66.7** | **36.4** | **16.6** | **83.3** | **56.0** | **39.4** | **63.6** | **64.4** |

## APPENDIX B: PERFORMANCE COMPARISON ON NUSCENES DATASET

To further evaluate the effectiveness of LiDAR-PTQ, we also conducted experiments on nuScenes (Caesar et al., 2020) dataset. Our performance evaluation involves two metrics, average precision (mAP) and nuScenes detection score (NDS). NDS is a weighted average of mAP and other attributes metrics, including translation, scale, orientation, velocity, and other box attributes. As shown in Tab 6, LiDAR-PTQ achieves state-of-the-art performance and outperforms BRECQ and QDrop by a large margin of 6.2 mAP and 3.9 mAP on CenterPoint-Pillar model and 5.3 mAP and 2.5 mAP on CenterPoint-Voxel model. Consistent with the accuracy on the Waymo dataset, our LiDAR-PTQ also achieves almost the same performance as the full precision model on nuScenes dataset.

## APPENDIX C: LiDAR-PTQ FOR POINT CLOUD SEGMENTATION

Table 7: The PTQ performance of SPVNAS on SemanticKITTI *val* set.

| Method | mIoU | car | bicycle | motorcycle | truck | other-vehicle | person | bicyclist | motorcyclist | road | parking | sidewalk | other-ground | building | fence | vegetation | trunk | terrain | pole | traffic sign |
|---|---|---|---|---|---|---|---|---|---|---|---|---|---|---|---|---|---|---|---|---|
| Full Prec. | 65.0 | 96.3 | 49.0 | 77.6 | 74.4 | 51.8 | 75.2 | 88.2 | 5.7 | 93.4 | 44.6 | 81.0 | 3.5 | 89.5 | 56.5 | 87.8 | 68.4 | 75.1 | 67.1 | 49.6 |
| Entropy | 46.9 | 92.9 | 34.7 | 72.1 | 20.4 | 37.2 | 48.5 | 80.9 | 5.1 | 47.8 | 16.9 | 28.7 | 0.2 | 79.9 | 47.5 | 82.9 | 57.0 | 44.0 | 55.9 | 38.8 |
| Max-min | 62.4 | 94.5 | 46.2 | 75.3 | 73.0 | 50.2 | 73.6 | 86.4 | 5.7 | 92.3 | 41.5 | 78.9 | 2.1 | 87.3 | 53.4 | 85.1 | 65.3 | 71.8 | 63.0 | 48.6 |
| **LiDAR-PTQ** | **64.9** | **96.3** | **48.7** | **78.0** | **74.3** | **52.2** | **74.5** | **87.9** | **5.9** | **93.3** | **44.0** | **80.9** | **3.5** | **89.4** | **56.4** | **87.6** | **68.3** | **74.5** | **67.2** | **49.5** |

Additionally, we conducted experiments on SemanticKITTI (Behley et al., 2019) dataset for point cloud segmentation to further evaluate the generalization of LiDAR-PTQ. Specifically, we utilize SPVNAS (Tang et al., 2020) as our baseline, which is a representative work in point cloud segmentation task. As shown in Tab 7, adopting entropy calibration leads to a significant accuracy drop of **18.09 mIOU**. As for a vanilla max-min calibration, there is still a performance drop **2.64 mIOU**

for quantized SPVNAS. However, LiDAR-PTQ can further achieve comparable accuracy to its float counterpart. This demonstrates the effectiveness of LiDAR-PTQ on point cloud segmentation tasks as well.

## APPENDIX D: EXPERIEMNTS DETAILS

**Dataset.** NuScenes dataset (Caesar et al., 2020) uses a LiDAR with 32 lines to collect data, containing 1000 scenes with 700, 150, and 150 scenes for training, validation, and testing, respectively. The metrics of the 3D detection task are mean Average Precision (mAP) and the nuScenes detection score (NDS). Waymo Open Dataset  (Sun et al., 2020) uses a LiDAR with 64 beams to collect data, containing 1150 sequences in total, 798 for training, 202 for validation, and 150 for testing. The metrics of the 3D detection task are mAP and mAPH (mAP weighted by heading). In Waymo, LEVEL1 and LEVEL2 are two difficulty levels corresponding to boxes with more than five LiDAR points and boxes with at least one LiDAR point. The detection range in nuScenes and WOD is 50 meters (cover area of 100m × 100m) and 75 meters (cover area of 150m × 150m).

**Implementation Details.** All the FP models in our paper use CenterPoint(Yin et al., 2021) official open-source codes based on Det3D (Zhu et al., 2019) framework. In WOD dataset, we randomly sample 256 frames point cloud data from the training set as the calibration data. The calibration set proportions is **0.16%** (256/158,081) for WOD. In nuScenes dataset, the calibration set proportions are **0.91%** (256/28,130). We set the first and the last layer of the network to keep full precision. We execute block reconstruction for the backbone and layer reconstruction for the neck and the head with a batch size of 4, respectively. Note that we do not consider using Int8 quantization for the PFN in CenterPoint-Pillar, since the input is 3D coordinates, with approximate range $\pm 10^2$ m and accuracy 0.01 m, so that Int8 quantization in FPN would result in a significant loss of information. The learning rate for the activation quantization scaling factor is 5e-5, and for weight quantization rounding, the learning rate is 5e-3. In TGPL loss, we set $\gamma$ as 0.1, and K as 500. We execute all experiments on a single Nvidia Tesla V100 GPU. For the speed test, the inference time of all comparison methods is measured on an NVIDIA Jeston AGX Orin, a resource-constrained edge GPU platform widely used in real-world autonomous driving.

## APPENDIX E: ENTROPY CALIBRATION METHOD

Given the original and quantized data distribution $p(i)$ and $q(i)$ as follows:

$$D_{KL}(p(i), q(i)) = \sum_i p(i) \log p(i) - p(i) \log q(i) \tag{13}$$

The entropy calibration method in Algorithm3

---

**Algorithm 2** Entropy calibration method

---

**Input**: FP32 histogram H with $N$ bins, and bit-width $b$.
**Output**: threshold with $min(D_{KL}(p(i), q(i)))$.
**Require:** $len(p) = len(q)$
 1: **for** $i$ in range($2^{b-1}$, $N$) **do**
 2:     ref_dist_p(i) = [bin[0], ...,bin[$i-1$]]
 3:     outliers_count = $sum$(bin[$i$],bin[$i+1$], . . . ,bin[$N-1$])
 4:     ref_dist_p(i)[$i-1$]+ = outliers_count
 5:     $p(i)$ =ref_dist_p(i)/$sum$(ref_dist_p(i))
 6:     quantize candidate_dist_q(i) from [ bin[0], . . . , bin[$i-1$]] into $2^{b-1}$ levels
 7:     candidate_dist_q(i)=$interp1d$((bin[0], ...,bin[127]), (bin[0], ...,bin[$i-1$]),method=$'linear'$)
 8:     $q(i)$ =candidate_dist_q(i)/$sum$(candidate_dist_q(i))
 9:     divergence[$i$] = $D_{KL}(p(i), q(i))$ using Eq 13
10: **end for**
11: $m = \text{argmin}\left(D = \left[\text{divergence}[2^{b-1}-1], ..., \text{divergence}[N-1]\right]\right)$
12: threshold = $(m + 0.5) * (width_{bin})$
13: **return**  threshold

---

## Appendix F: Gird Search

For a weight or activation tensor $X$, we can get their initial quantization scale factor using the following equation:

$$\hat{x} = (clamp(\lfloor \frac{x}{s} \rceil + z, q_{min}, q_{max}) - z) \cdot s \tag{14}$$

$$s = (x_{max} - x_{min}) / (2^b - 1) \tag{15}$$

$$\arg\min_{s_t} \|(X - \hat{X}(s_l))\|_F^2 \tag{16}$$

$\| \cdot \|_F^2$ is the Frobenius norm (MSE Loss). Refer to appendix for more details about grid search. Then linearly divide the interval $[\alpha s_0, \beta s_0]$ into $T$ candidate bins, denoted as $\{s_t\}_{t=1}^T$. $\alpha$, $\beta$ and $T$ are designed to control the search range and granularity. Finally, search $\{s_t\}_{t=1}^T$ to find the optimal $s_{opt}$ that minimizes the quantization error, The entropy calibration method in Algorithm3

---

**Algorithm 3** Grid search

---

**Input**: the input of full precision tensor $X$, bit-width $b$ and $T$ bins.
**Output**: scale factor $s_{opt}$ with $min(\|(X - \hat{X}(s_l))\|_F^2)$.
1: using $x_{max} = max(|x|)$ get max value of tensor $X$
2: set $range = x_{max}$, $c_{best} = 100$
3: set $v_{min} = x_{min}$ and $v_{max} = x_{max}$
4: **for** $i$ in range$(1, T)$ **do**
5:     threshold $= range/T/i$
6:     $x_{min} = -threshold$, $x_{max} = threshold$
7:     get scale $s_t$ with $x_{min}$ and $x_{max}$ using Eq 15
8:     input the quantized value $\hat{x}$ and FP value $x$ using Eq 14 to get score $c$
9:     update $v_{min}$ and $v_{max}$ when $c < c_{best}$ and update $c_{best} = c$
10: **end for**
11: get $v_{min}$ and $v_{max}$ with the minimal score $c$
12: get final scale $s_{opt}$ with $v_{min}$ and $v_{max}$ using Eq 15
13: **return** scale $s_{opt}$

---

