# OpenReview forum: "LiDAR-PTQ: Post-Training Quantization for Point Cloud 3D Object Detection"
_ICLR.cc/2024/Conference — ICLR 2024 poster_

### Official Review · Reviewer_fLuR · 2023-10-28

**Soundness:** 2 fair
**Presentation:** 3 good
**Contribution:** 2 fair
**Rating:** 6
**Confidence:** 4

**Summary:**

This paper proposes a post-training quantization method for LiDAR-based 3D object detection. It uses min-max quantization and proposes task-guided global positive loss.
The experiments are conducted in the Waymo Open Dataset with CenterPoint detector, achieving competitive performance among other quantization methods.

**Strengths:**

- This paper achieves competitive results compared with other quantization methods.
- The writing is clear and easy to follow.
- Although the proposed method is quite simple, it is fast and pratical.

**Weaknesses:**

### This paper lacks novelty and originality.
- Although the authors propose the so-called sparsity-based calibration and conduct the spatial sparsity analysis, the implementation simply adopts the max-min calibration without any unique designs.
- The proposed TGPL is also a trivial technique, which basically selects some high-confidence prediction for supervision and this is widely-adopted in many fields such as distillation.
### Some claims are inaccurate or wrong.
"Point cloud coordinates have explicitly encoded the shapes and geometrics of objects, so the larger arithmetic range of input point
cloud preserves essential geometric information."  By this sentence, the authors indicate the large range of point cloud coordinates makes the detectors sensitive to quantization range. However, in fact, LiDAR-based object detectors do not take the absolute point cloud coordinates as input. Voxel encoders usually calculate the relative coordinates within each voxel as input.
### The application is a little bit narrow.
There are emerging methods of "fully sparse detectors[1][2][3]", which show impressive efficacy and efficiency and do not lead to zero feature maps. Thus the proposed method (sparsity-based calibration) has no advantages in these methods.

[1] Fully Sparse 3D Object Detection, NeurIPS 2022. \
[2] Super Sparse 3D Object Detection, TPAMI 2023. \
[3] VoxelNeXt: Fully Sparse VoxelNet for 3D Object Detection and Tracking, CVPR 2023.

**Questions:**

No more additional questions. Looking forward to the authors' response to my concerns in the weaknesses box.

---

> ### Author Response · Authors · 2023-11-17
> **Effectiveness on fully sparse detector**
>
> **Q1**: Defend our novelty.
>
> **A1**:
> Our research is rooted in a deep understanding of the sparsity characteristics in the LiDAR scenario. We discover two fundamental phenomena (i and ii on Page 5) when it comes to quantization over point cloud data that intrinsically bears sparsity. The choice of the max-min calibration strategy is non-trivial. By selecting maximum/minimum values, we are free from the influence of sparse data distribution. It retains the most effective information and is not affected by the large number of zeros. On the contrary, an entropy-based calibration seeks to preserve data distribution which suffers from abundant zero values and sparse information, rendering much inferior performance (Table 1 and 2). For instance, when the distance range gets beyond 50m, its mAPH suddenly drops from 38.09 to 5.9. Besides, Our proposed max-min doesn't come alone, we apply a simple grid search strategy for better quantization scales. As for TGPL, we pursue a global supervised mechanism to bridge the performance gap caused by the previous per-layer quantization. These three components altogether bring a state-of-the-art performance shown in Table 4. Each component alone may seem simple, but an orchestration of them requires deeper thoughts.
>
> **Q2**: Claims about point cloud coordinates.
>
> **A2**: In fact, absolute coordinates are also used in practice. In BEV-based point cloud detectors, such as voxel feature encoding (VFE) and pillar feature encoding (PFE), **the Euclidean distance** calculated based on the absolute coordinate of the point cloud or the absolute coordinates are also utilized as input features, along with the relative coordinates of voxels or pillars,  which is a common operation. Please refer to open-source code \url{https://github.com/tianweiy/CenterPoint/blob/d3a248fa56db2601860d576d5934d00fee9916eb/det3d/models/readers/pillar_encoder.py#L147} or \url{https://github.com/open-mmlab/mmdetection3d/blob/5c0613be29bd2e51771ec5e046d89ba3089887c7/mmdet3d/models/voxel_encoders/voxel_encoder.py#L432}.
>
> **Q3:** Application on other LiDAR detectors.
>
> **A3:** Thanks for the advice about more LiDAR detectors. We have conducted experiments on FSD, an excellent fully sparse 3D detector, as shown in the table below. Adopting entropy calibration still leads to a significant accuracy drop of -61.50 mAPH/L2. We discover that quantized FSD readily delivers the desired performance while employing a vanilla max-min calibration. Nonetheless, using LiDAR-PTQ can further achieve comparable accuracy to its float counterpart. This demonstrates the effectiveness of LiDAR-PTQ on fully-sparse detector as well. To our best knowledge, the mainstream approaches in industrial applications are still non-fully sparse detectors like VoxelNet[1], PointPillars[2], and CenterPoint[3] because of friendly deployment using off-the-shelf frameworks like TensorRT. We emphasize that our method can be seamlessly integrated to enjoy a significant speed-up, which is vital for the community. In summary, our LiDAR-PTQ is not only limited to specific BEV-based point cloud detectors. Given such generalization ability and practical value for promoting the application of 3D point cloud detectors on edge devices, we believe our method deserves to be shared with our community.
> | Models | Methods    | Bits(W/A) | Mean mAP/L2 | Mean  mAPH/L2 | Veh mAP/L2 | Veh mAPH/L2 | Ped mAP/L2 | Ped mAPH/L2 |
> |--------|------------|-----------|---------------|----------------|--------------|---------------|--------------|---------------|
> | FSD | Full Prec. | 32/32     | 73.01         | 70.94          | 70.34        | 69.98         | 73.95        | 69.16         |
> | FSD | Entropy    | 8/8       | 10.54 (**-62.47**)| 9.44 (**-61.50**)  | 0.06         | 0.06          | 21.88        | 18.86         |
> | FSD  | Max-min    | 8/8       | 71.24 (**-1.77**) | 69.37 (**-1.57**)  | 68.42        | 68.18         | 72.09        | 67.60        |
> | FSD  | LiDAR-PTQ  | 8/8       | **72.84**     | **70.73**      | **69.95**    | **69.62**     | **73.85**    | **68.95**     |
>
> References:
>  - [1] VoxelNet: End-to-end learning for point cloud based 3D object detection. CVPR 2018.
>  - [2] Pointpillars: Fast encoders for object detection from point clouds. CVPR 2019.
>  - [3] CenterPoint: Center-based {3d} object detection and tracking. CVPR 2021.

---

> > ### Comment · Reviewer_fLuR · 2023-11-20
> >
> > Thanks for the additional experiments, which suggest the proposed method is also suitable for the fully sparse detectors.
> > It seems to be a good method in industrial, so I suggest authors release the source code, making the work more valuable.
> > There is another minor issue. Although the authors point out that many methods can adopt the absolute coordinates, but, in fact, this functionality is usually not enabled. See https://github.com/open-mmlab/mmdetection3d/blob/5c0613be29bd2e51771ec5e046d89ba3089887c7/configs/_base_/models/pointpillars_hv_secfpn_waymo.py#L21
> >
> > The additional experiments should be incorporated into the revised paper.

---

> > > ### Author Response · Authors · 2023-11-20
> > > **Our code will be released.**
> > >
> > > Dear reviewer:
> > >
> > > Many thanks for your valuable remarks.
> > >
> > >
> > > ***Q1:*** Code and additional experiments.
> > >
> > > ***A1:***
> > >  Thanks for your advice. Regarding the code, we will release it to the community soon. About the additional experiments, we have incorporated them in the supplementary materials and have added more details in the revised paper.
> > >
> > > ***Q2:*** Make claims more accurate.
> > >
> > > ***A2:***
> > > We have updated the statement of point cloud encoding in Sec 3.2. in the revised paper. (changes are marked with blue).
> > >
> > > We hope to resolve your doubts with our best efforts. Please share your thoughts on viewing our reply.
> > >
> > > Regards,
> > >
> > > The authors

---

### Official Review · Reviewer_3wrT · 2023-11-03

**Soundness:** 3 good
**Presentation:** 3 good
**Contribution:** 3 good
**Rating:** 6
**Confidence:** 3

**Summary:**

Running neural nets for object detection on an edge device requires quantization of the network so that it can fit on the limited memory of the device and perform inference at an adequate (real-time) speed.
Traditionally, post-training quantization (PTQ) for vision tasks utilizes a small set of unlabeled samples as a calibration set. These samples are used to quantize the weights and activations of the network using max-min and information theory to minimize KL-divergence between the quantized and non-quantized tensor. The information-theory based method which performs well for 2D object detection does quite poorly on 3D LIDAR object detection because of sparsity in the point cloud. This paper describes a LIDAR-PTQ method that quantizes existing LIDAR object detectors like the Centre-point-pillar. It uses a sparsity based calibrator, a Task-Guided Loss (which calibrates the quantization parameters using performance on the final object detection task) and an adaptive rounding-to-nearest loss which determines whether the weight in a particular neuron should be rounded up or down.

Experiments demonstrate similar performance to the floating point model on the NuScenes and Waymo datasets, but with a 3x inference speedup on a Jetson AGX Orin edge device.

**Strengths:**

The paper is on the whole well written and does a good job of introducing the reader to the subject of neural network quantization. It describes an algorithm for the important task of adapting large neural nets for operation on the edge, while preserving model accuracy. Since there is relatively little work on LIDAR network quantization, the paper is timely and important.

**Weaknesses:**

There is not enough detail about the optimization steps: particularly which parameters are optimized when and what percentage of training data is required for this post-training quantization. The task-guided loss minimizes the distance between the output of the quantized network vs the non-quantized network and so presumably requires the labelled training data. This seems to deviate from the requirements of PTQ, which only seems to need a small number of unlabeled samples as the calibration set.
Even though the combined system with max-min quantization + grid search + TGPL + Adaptive Rounding together approach the mAP of the FP model, the contribution in narrowing the gap between max-min and the final model seems to be mainly by the grid search. The TGPL and Adaptive Rounding contribute relatively small gains to the mAP. This could be a quirk of the dataset / optimization parameters / training regime and I would ideally like to see more experiments to be convinced of the efficacy of TGPL and Adaptive Rounding, even though from a conceptual point of view, they make sense.
I would like the code to be released so that independent validation of the authors’ experiments can be performed.

**Questions:**

-What is LEVEL_2 mAPH in Table 1? Please clarify this closer to the table
-Page 5: Huge ‘sparisity’ lead to inappropriate quantization range.
‘Sparisity’ should be sparsity
-Page 7: ‘… are the soft-quantized weights follows Eq 11’.
I think you mean equation 10?
-Page 7: ‘NuSecens’ should be NuScenes

---

> ### Author Response · Authors · 2023-11-17
> **More training details**
>
> **Q1**: Training details
>
> **A1**: In LiDAR-PTQ, for every quantized layer, the optimized parameter is weight scale $s_w$, weight zero-point $z_w$, activation scale $s_a$, activation zero-point $z_a$  and adaptive rounding value for weight $\theta$. We generally summarize the overall optimization step as follows: in every quantized layer
>
> 1.  using sparsity-based calibration to get initial $s_w$, $z_w$, $s_a$, $z_a$ and $\theta$.
>
>  2.  using ${L}_{total}$ to fine-tune the $s_w$, $z_w$, $s_a$, $z_a$ and $\theta$.
>
> The above steps are calculated for every quantized layer in quantized model. More details refer to Algorithm 1 in Sec 3.2.
>
> **Q2**: Calibrated data percentage.
>
> **A2**: In WOD dataset, we randomly sample 256 frames point cloud data from the training set as the calibration set. The calibration set proportions is 0.16% (256/158,081) for WOD. Notably, we do not use the ground truth labels for these point cloud data due to we are PTQ method.
>
> **Q3**: Presumably requires the labelled training data.
>
> **A3**: We do not require any labeled data to train the quantized model. We use the prediction of FP model to supervise the prediction of quantized model. Employing a method similar to distillation to enhance the quantized model using its float counterpart is effective in PTQ scenarios.
>
> **Q4**: Gains of TGPL and Adaptive round.
>
> **A4**: The contribution of TGPL and adaptive rounding is to mitigate the final performance gap and achieve almost the same performance as FP model. There are many advanced 2D PTQ methods (like BRECQ[1], QDROP[2]) that can be used in point cloud detection models quantization. However, they still suffer from performance drops (see Tab 3). In this paper, our motivation is to propose a PTQ method designed for point cloud detection tasks and achieve comparable accuracy with FP model. Although the proposed sparsity-based calibration has shown huge performance gain, there still exists a significant gap compared with the FP model. Therefore, we propose TGPL and adaptive rounding to narrow the final performance gap.
>
> **Q5**: Open-sourced code.
>
> **A5**: We are arranging the code and will release it to the community later.
>
> **Q6**: LEVEL2 in Waymo dataset:
>
> **A6**: In Waymo Open Dataset (WOD), each ground truth label is categorized into different difficulty levels:
>    - LEVEL1 (L1): if number of LiDAR points $n$ $\textgreater$ 5 in ground truth boxes.
>    - LEVEL2 (L2): if number of LiDAR points 1 $\leq n \leq$ 5 in ground truth boxes.
> When evaluating, L2 metrics are computed by considering both L1 and L2 ground truth. The Waymo official leaderboard ranks based on Mean Average Precision with Heading / LEVEL2 (mAPH/L2) metrics, which is also widely adopted metrics by the community. We have added more details in the manuscript to make it clearer.
>
> **Q7**: Typos:
>
> **A7**: We apologize for our error. We have corrected all the typos carefully in the manuscript.
>
> **Q8**: Page 7, Eq 11:
>
> **A8**: Yes. In Sec 3.4, the soft-quantized weights is calculated by Eq 10 and Eq 2, and then we calculate ${L}_{total}$ using Eq 11.
>
> References:
>  * [1] Brecq: Pushing the limit of post-training quantization by block reconstruction. ICLR 2021.
>  * [2] QDrop: Randomly Dropping Quantization for Extremely Low-bit Post-Training Quantization. ICLR 2022.

---

> ### Author Response · Authors · 2023-11-23
> **Sincerely looking forward to your feedbacks**
>
> Dear Reviewer 3wrT,
>
> Thanks again for the reviews. We expect any further feedback to enhance our paper and address your rest concerns before the deadline. We are desirous to have a productive conversation.
>
> Sincerely,
>
> The Authors

---

### Official Review · Reviewer_Dy1B · 2023-11-07

**Soundness:** 3 good
**Presentation:** 3 good
**Contribution:** 3 good
**Rating:** 6
**Confidence:** 3

**Summary:**

This paper proposes a Post-Training Quantization (PTQ) compression approach, LiDAR-PTQ, for the real applications of 3D LiDAR point cloud detection that are deployed on edge GPU systems. The key ideas include density-aware calibration, task-guided global positive loss and layerwise-reconstruction-based adaptive rounding.  The proposed method is evaluated on the Waymo dataset to validate the low performance-drop with INT8 compared with FP32.

**Strengths:**

- The paper is  easy to understand.  The analysis of why 2D-PTQ can’t be directly applied to 3D is reasonable and the corresponding solution is practical.
-  The proposed LiDAR-PTQ achieves competitive  detection performance as INT8 quantization for bothe CP-Pillar and Voxel.

**Weaknesses:**

- The method is only evalutated on the PointCenter model, how about the effect for other types of detectors (Transform-based, etc.).
- It is said that the method is evaluated on various datasets (Waymo and NuScenes). However, results on  NuScenes seem not available.

**Questions:**

- Comparisons of inference acceleration performance with other baselines are not available in Tab.3.  Accuracy together with speed performance can better demonstrate the advantages of the proposed method.
- How is γ set for TGPL?
- It is wrotten that “there is no extra access to labeled training data”. Dose this work like the knowledge distillaatio  from the full FP32 model  for an INT8 quantizated model?
- For results on Waymo, can authors give some explaination about why  select LEVEL2?

---

> ### Author Response · Authors · 2023-11-17
> **Add more implementation details**
>
> **Q1**: Add inference performance in Tab.3
>
> **A1**: Thanks, we have updated it in Tab. 3.
>
> **Q2**: γ setting in TGPL
>
> **A2**: In TGPL loss, we set γ as 0.1 to filter predictions from the FP model, and then we select the top K boxes, where we set K as 500. We have updated in manuscript.
>
> **Q3**: This work like the knowledge distillation from FP model for quantized model?
>
> **A3**: Yes, employing a method similar to distillation to enhance the quantized model using its float counterpart is effective in PTQ scenarios. To further improve our approach, we have introduced a global distillation scheme called TGPL. Different from the commonly used knowledge distillation paradigm, we did not use any labels of the calibration set in the whole quantization process in LiDAR-PTQ.
>
> **Q4**: LEVEL_2 (L2) in Waymo dataset
>
> **A4**: In Waymo Open Dataset (WOD), each ground truth label is categorized into different difficulty levels.:
>    - LEVEL1 (L1): if number of LiDAR points $n$ $\textgreater$ 5 in ground truth boxes.
>    - LEVEL2 (L2): if number of LiDAR points 1 $\leq n \leq$ 5 in ground truth boxes.
>
> When evaluating, L2 metrics are computed by considering both L1 and L2 ground truth. **The Waymo official leaderboard ranks based on Mean Average Precision with Heading / LEVEL2 (mAPH/L2) metrics**, which is also widely adopted metrics by the community.
>
> **Q5**: Effect for other types of detectors (such as Transform-based).
>
> **A5**: 1. Thank for your suggestions. Currently, this work primarily focuses on CNN-based Lidar detectors, which remain the mainstream approach in current industrial applications. Besides, we endorse the potential and promising prospects of Transformer-based methods. Combination of Transformer networks and Lidar data will bring further intriguing challenges. We are eager to explore this topic in our future works.
>
> 2.  Not only limited to the CenterPoint (-Pillar and -Voxel) model, we also evaluated the effectiveness of our method on fully sparse detectors, such as FSD[1], and achieved comparable performance with the FP model. Additionally, we provided the differences between different model architectures, which show LiDAR-PTQ is applicable to differnt cnn-based detectors.
> | Models    | Methods    | Bits(W/A) | Mean mAP/L1 | Mean mAPH/L2 | Veh mAP/L2 | Veh  mAPH/L2 | Ped mAP/L2 | Ped mAPH/L2 |
> |-----------|------------|-----------|---------------|----------------|--------------|---------------|--------------|---------------|
> | FSD[1]   | Full Prec. | 32/32     | 73.01         | 70.94          | 70.34        | 69.98         | 73.95        | 69.16         |
> | FSD[1]   | LiDAR-PTQ  | 8/8       | **72.84**     | **70.73**      | **69.95**    | **69.62**     | **73.85**    | **68.95**     |
>
> | Method    | Representation | Backbone | Neck   | Head   |
> |-----------|----------------|----------|--------|--------|
> | CP-Pillar | Pillar         | Dense    | Dense  | Dense  |
> | CP-Voxel  | Voxel          | Sparse   | Dense  | Dense  |
> | FSD       | Point+Voxel    | Sparse   | Sparse | Sparse |
>
> References:
> *  [1]. Fully Sparse 3D Object Detection, NeurIPS 2022.

---

> ### Author Response · Authors · 2023-11-21
> **Update experiments on nuScenes dataset**
>
> ***Q1:*** nuScenes dataset experiments
>
> ***A1:***  Thank you for your reminder. In the revision, we have updated the performance on the nuScenes [1] dataset. In the nuScenes dataset, we randomly sample 256 frames point cloud data from the training set as the calibration set. The calibration set proportions are 0.91\% (256/28,130). Our performance evaluation involves two metrics, average precision (mAP) and nuScenes detection score (NDS). NDS is a weighted average of mAP and other attributes metrics, including translation, scale, orientation, velocity, and other box attributes. As shown in the table below, LiDAR-PTQ sitll shows state-of-the-art performance, consistent with its accuracy on Waymo dataset. Our LiDAR-PTQ achieves almost the same performance as the full precision model on the nuScenes dataset. All experiments have been updated in the supplementary materials.
>
> | Models    | Methods     | Bits(W/A) | NDS  | mAP  | Car  | Truck | Bus  | Trailer | CV   | Ped  | Motor | BC   | TC   | Barrier |
> |-----------|-------------|-----------|------|------|------|-------|------|---------|------|------|-------|------|------|---------|
> | CP-Pillar | Full Prec.  | 32/32     | 60.3 | 50.0 | 83.8 | 50.6  | 61.8 | 31.2    | 9.2  | 79.4 | 44.1  | 20.2 | 57.7 | 61.3    |
> | CP-Pillar | BRECQ       | 8/8       | 56.9 | 43.6 | 75.9 | 41.4  | 54.3 | 21.6    | 3.8  | 78.1 | 37.4  | 15.7 | 55.0 | 53.3    |
> |CP-Pillar | QDrop       | 8/8       | 57.8 | 45.9 | 78.8 | 44.2  | 57.0 | 23.8    | 5.2  | 78.4 | 40.1  | 17.6 | 56.7 | 56.8    |
> | CP-Pillar| PD-Quant    | 8/8       | 59.6 | 48.3 | 81.8 | 47.6  | 59.4 | 28.2    | 7.8  | 78.4 | 41.6  | 19.8 | 57.6 | 61.0    |
> | CP-Pillar| LiDAR-PTQ   | 8/8       | ***60.2*** | ***49.8*** | 83.7 | 50.8  | 61.8 | 30.6    | 9.0  | 79.0 | 43.6  | 20.4 | 57.8 | 61.0    |
> | CP-Voxel  | Full Prec.  | 32/32     | 64.8 | 56.6 | 84.6 | 54.5  | 66.7 | 36.4    | 16.9 | 83.1 | 56.1  | 39.6 | 64.0 | 64.3    |
> | CP-Voxel  | BRECQ       | 8/8       | 62.0 | 51.2 | 76.5 | 46.8  | 60.5 | 28.9    | 12.5 | 80.4 | 53.7  | 34.8 | 58.8 | 59.1    |
> | CP-Voxel  | QDrop       | 8/8       | 63.2 | 54.0 | 82.1 | 48.5  | 64.9 | 32.9    | 15.1 | 81.1 | 55.1  | 36.9 | 60.9 | 63.7    |
> | CP-Voxel  | PD-Quant    | 8/8       | 63.7 | 55.2 | 83.7 | 51.1  | 66.6 | 34.1    | 16.6 | 82.8 | 55.1  | 36.4 | 62.6 | 63.2    |
> | CP-Voxel  | LiDAR-PTQ   | 8/8       | ***64.7*** | ***56.5*** | 84.6 | 54.2  | 66.7 | 36.4    | 16.6 | 83.3 | 56.0  | 39.4 | 63.6 | 64.4    |
>
> We hope to resolve your doubts with our best efforts. Please share your thoughts on viewing our reply.
>
> References:
> [1]  Nuscenes: A multimodal dataset for autonomous driving. CVPR, 2020.

---

> ### Author Response · Authors · 2023-11-23
> **Looking forward to hearing your response**
>
> Dear Reviewer Dy1B,
>
> Many thanks for your valuable remarks.
>
> To make the best use of the discussion period and to improve our work, we'd like to hear your response to know whether we have addressed your concerns or there are any new questions arise.
>
> We hope to resolve your doubts with our best efforts. Please share your thoughts on viewing our reply.
>
> Regards,
>
> The authors

---

### Official Review · Reviewer_szHU · 2023-11-09

**Soundness:** 3 good
**Presentation:** 3 good
**Contribution:** 2 fair
**Rating:** 6
**Confidence:** 4

**Summary:**

This paper describes an approach for model compression for applications using 3D Lidar data, namely 3D object detection.  The approach is based on post-training quantization.  The paper shows that direct application of post-training quantization methods used with 2D images leads to a decrease in performance. The method proposed in the paper is based on a sparsity-based calibration followed by the application of a task -guided global positive loss and finally by the application of an adaptive rounding-to-nearest operation. Experimental results show that the proposed approach achieves state-of-the-art performance, namely that the accuracy of the model with PTQ INT8 is at the level of the accuracy of the model using FP32. The inference speed is increased 3 times.

**Strengths:**

The paper shows that the simple and direct application of approaches commonly used for model compression in applications with 2D images leads to a decrease in performance (when applied to object detection with 3D lidar data) and proposes a new method for model compression (for 3D Lidar data and for object detection). This new model achieves similar accuracy to a full precision model and achieves an increase in inference speed (3x). The authors also analyze and discuss the reasons that may be the cause of such behavior. Such analysis is qualitative---no experimental validation. The proposed approach is clear and contributes a solution to a well-defined problem.

**Weaknesses:**

The main weakness of the paper results from the experimental results (and validation) having been obtained for a single application: CenterPoint (in both versions, Pillar and Voxel).  This is too specific, and it is not clear that similar results could be obtained with other applications using 3D Lidar data, e.g., segmentation.

**Questions:**

Two main questions:
--Would  similar results be obtained with other object detectors based on Lidar data?
--Would similar results be obtained with other applications using Lidar data, e.g., segmentation?

**Details Of Ethics Concerns:**

No ethics concerns.

---

> ### Author Response · Authors · 2023-11-17
> **Similar results is also obtained with other object detectors**
>
> **Q1**: Would be similar results on other object detectors?
>
> **A1**: 1. CenterPoint integrates two milestone works in LiDAR-based BEV detection, VoxelNet and PointPillars as CP-Pillar and CP-Voxel. In particular, CP-Pillar and CP-Voxel have different network designs. The CP-Pillar model is a fully dense convolutional network, while the CP-Voxel model includes SPConv and dense convolution. Our results on CenterPoint (-pillar and -voxel) demonstrate that:
>    * Lidar-PTQ is applicable to pillar-based and voxel-based detectors.
>    * Lidar-PTQ is applicable to SPConv and dense convolution operations.
> 2. Except CP-Pillar and CP-Voxel models, to evaluate the generalization ability of our method, we also evaluate our LiDAR-PTQ on FSD, a fully sparse 3D detector. The experiments demonstrate that our method is also applicable to fully sparse detectors.
>
> | Models    | Methods    | Bits(W/A) | Mean mAP/L2 | Mean mAPH/L2 | Veh mAP/L2 | Veh mAPH/L2 | Ped mAP/L2 | Ped mAPH/L2 |
> |-----------|------------|-----------|---------------|----------------|--------------|---------------|--------------|---------------|
> | CP-Pillar | Full Prec. | 32/32     | 65.78         | 60.32          | 65.92        | 65.42         | 65.65        | 55.23         |
> | CP-Pillar  | LiDAR-PTQ  | 8/8       | **65.60**     | **60.12**      | **65.64**    | **65.14**     | **65.55**    | **55.11**     |
> | CP-Voxel  | Full Prec. | 32/32     | 67.67         | 65.25          | 66.29        | 65.79         | 68.04        | 62.35         |
> | CP-Voxel  | LiDAR-PTQ  | 8/8       | **67.60**     | **65.18**      | **66.27**    | **65.78**     | **67.95**    | **62.24**     |
> | FSD       | Full Prec. | 32/32     | 73.01         | 70.94          | 70.34        | 69.98         | 73.95        | 69.16         |
> | FSD       | LiDAR-PTQ  | 8/8       | **72.84**     | **70.73**      | **69.95**    | **69.62**     | **73.85**    | **68.95**     |
>
> | Models    | Representation  | Backbone  | Neck | Head |
> |-----------|------------|-----------|---------------|---------------|
> | CP-Pillar | Pillar | dense     | dense        |dense    |
> | CP-Voxel | Voxel | sparse     | dense      |dense |
> | FSD       | Point+Voxel | sparse     | sparse      |sparse |
>
> **Q2**: Would be similar results on other object applications, e.g., segmentation?
>
> **A2**: Thanks for your constructive suggestions.
> 1. For a timely response, we have not enough time to provide experiments for point cloud segmentation task due to the original intention of this paper is to focus on the LiDAR-based BEV detection task. LiDAR point cloud segmentation is another topic, which beyond the original scope of this study.
> 2. Furthermore, we are actively conducting experiments in point cloud segmentation task, and if time permits, we will provide the related results before the rebuttal deadline to better evaluate our LiDAR-PTQ method.

---

> ### Author Response · Authors · 2023-11-20
> **Similar results is also obtained on  point cloud segmentation applications**
>
> ***Q1:*** Would be similar results on other applications using Lidar data, e.g., segmentation.
>
> ***A1:*** Given your prior constructive suggestions, we conducted experiments on SemanticKITTI [1] dataset for point cloud segmentation to further evaluate the generalization of LiDAR-PTQ. Specifically, we utilize SPVNAS [2] as our baseline, which is a representative work in point cloud segmentation. As shown in table below, adopting entropy calibration leads to a significant accuracy drop of ***18.09 mIOU***. As for a vanilla max-min calibration, there is still a performance drop ***2.64 mIOU*** for quantized SPVNAS. However, LiDAR-PTQ can further achieve comparable accuracy to its float counterpart. This demonstrates the effectiveness of LiDAR-PTQ on point cloud segmentation tasks as well. For more detailed performance of different classes refer to Tab. 2.
>
> | Models       | Methods    | Bits(W/A) | mIOU         |
> |--------------|------------|-----------|--------------|
> |              | Full Prec. | 32/32     | 65.01        |
> | SPVNAS-Tiny  | Entropy    | 8/8       | 46.92 (-18.09)|
> |              | Max-min    | 8/8       | 62.37 (-2.64) |
> |              | **LiDAR-PTQ** | 8/8   | **64.86**    |
>
> | Method     | mIoU | car  | bicycle | motorcycle | truck | other-vehicle | person | bicyclist | motorcyclist | road | parking | sidewalk | other-ground | building | fence | vegetation | trunk | terrain | pole | traffic sign |
> |------------|------|------|---------|------------|-------|---------------|--------|-----------|--------------|------|---------|----------|--------------|----------|-------|------------|-------|---------|------|--------------|
> | Full Prec. | 65.0 | 96.3 | 49.0    | 77.6       | 74.4  | 51.8          | 75.2   | 88.2      | 5.7         | 93.4 | 44.6    | 81.0     | 3.5          | 89.5     | 56.5  | 87.8       | 68.4  | 75.1    | 67.1 | 49.6         |
> | Entropy    | 46.9 | 92.9 | 34.7    | 72.1       | 20.4  | 37.2          | 48.5   | 80.9      | 51.5        | 47.8 | 16.9    | 28.7     | 0.2          | 79.9     | 47.5  | 82.9       | 57.0  | 44.0    | 55.9 | 38.8         |
> | Max-min    | 62.4 | 94.5 | 46.2    | 75.3       | 73.0  | 50.2          | 73.6   | 86.39     | 57.2        | 92.3 | 41.5    | 78.9     | 2.1          | 87.3     | 53.4  | 85.1       | 65.3  | 71.8    | 63.0 | 48.6         |
> | LiDAR-PTQ  | **64.9** | 96.3 | 48.7    | 78.0       | 74.3  | 52.2          | 74.5   | 87.9      | 5.9         | 93.3 | 44.0    | 80.9     | 3.5          | 89.4     | 56.4  | 87.6       | 68.3  | 74.5    | 67.2 | 49.5         |
>
>
>
> We hope to resolve your doubts with our best efforts. Please share your thoughts on viewing our reply.
>
>
> References:
>   1. SemanticKITTI: A dataset for semantic scene understanding of lidar sequence, ICCV 2019.
>   2. Searching efficient 3d architectures with sparse point-voxel convolution, ECCV 2020.

---

> > ### Comment · Reviewer_szHU · 2023-11-23
> > **Thank you for your answers and additional results for segmentation.**
> >
> > Your answers clarified the issues related with object detectors. You have also added additional results specifically for segmentation (which was not the focus of the paper) ---thank you for the additional results. These additional results clearly validate your approach and therefore I think that the contribution of your approach is solid and clear.

---

### Author Response · Authors · 2023-11-17
**General response and revision update notice**

Dear reviewers,

We thank everyone's effort for the precious review. The paper has been moderately revised according to the suggestions received. We've carefully examined all the questions to write the answers. Please feel encouraged to discuss actively with us if any new doubts occur. All changes in manuscript are marked with blue.

Here is a list of updates (by page order) in the paper and appendix:
   - Revised the description of LEVEL\_2 mAPH in Tab.1 (suggested by Reviewer Dy1B and 3wrT) to make it clearer. (Section 3.1)
   - Added more algorithm details in  Algorithm 1 to make it clearer. (Section 3.1)
   - Added relative performance drop in Table 2 to make it clearer. (Section 3.2)
   - Revised the sentence at the end of page 6 (suggested by Reviewer 3wrT) to make it clearer. (Section 3.4)
   - Added inference acceleration performance with other baselines in Tab.3 (suggested by Reviewer Dy1B). (Section 4.1)
   - Added more implementation details at the beginning of page 7 (suggested by Reviewer Dy1B and 3wrT) to make it clearer. (Section 5.1)
   - Fixed typos mentioned by Reviewer 3wrT of Page 5 and page 7. (Section 5.2)
   - Added experiments on other type detectors (fully sparse detector) on Sup.

Sincerely,

The Authors

---

> ### Comment · Area_Chair_GwPo · 2023-11-19
> **Please read the authors' responses**
>
> Dear reviewers,
>
> Could you please read the authors' responses and give your feed back? The period of Author-Reviewer Discussions is Fri, Nov 10 – Wed, Nov 22.
>
> Many thanks,
>
> AC

---

### Meta-Review · Area_Chair_GwPo · 2023-12-05

**Metareview:**

This paper proposes an effective Post-Training Quantization (PTQ) method called LiDAR-PTQ. The proposed method is particularly curated for 3D lidar detection. Extensive experimental results confirm that the LiDAR-PTQ achieves state-of-the-art quantization performance when applied to CenterPoint. And the LiDAR-PTQ is cost-effective.

This paper receives four “marginally above the acceptance threshold” ratings.

Reviewer szHU gives “marginally above the acceptance threshold”. Reviewer szHU is satisfied with the performance and the efficiency of the proposed method. Reviewer szHU thinks the proposed approach is clear. But also,  Reviewer szHU thinks the experiments are too application-specific.  The authors gave their rebuttals. After reading them, Reviewer szHU has no further questions.

Reviewer Dy1B gives “marginally above the acceptance threshold”. Reviewer  Dy1B thinks this paper is easy to understand.  Reviewer  Dy1B is satisfied with the performance of the proposed method. But also, Reviewer Dy1B wants to see the effect for other types of detectors.  Reviewer Dy1B points out that the results on NuScenes seem not available.  The authors gave their rebuttals. Reviewer  Dy1B still gives “marginally above the acceptance threshold”.

Reviewer 3wrT gives “marginally above the acceptance threshold”. Reviewer 3wrT thinks this paper is well-written. Since there is relatively little work on LIDAR network quantization, Reviewer 3wrT thinks the paper is timely and important. But also, Reviewer 3wrT wants to see more details about optimization steps. The authors gave their rebuttals. Reviewer  3wrT still gives “marginally above the acceptance threshold”.

Reviewer fLuR gives “marginally above the acceptance threshold”. Reviewer fLuR thinks this paper achieves competitive results compared with other quantization methods. Reviewer fLuR thinks the writing is clear. Reviewer fLuR thinks the proposed method is practical. But also, Reviewer fLuR thinks the paper lacks novelty.  Reviewer fLuR thinks some claims are inaccurate. The authors’ rebuttals partially address the concerns of Reviewer fLuR.

Therefore, based on the reviewers’ comments, the paper can be accepted by ICLR.

**Justification For Why Not Higher Score:**

The idea of this paper is not very novel. Additional experiments mentioned by reviewers should be incorporated into the revised paper.

**Justification For Why Not Lower Score:**

The proposed method is practical. Both the performance and the efficiency of the proposed method are satisfactory.

---

### Decision · Program_Chairs · 2024-01-16

Accept (poster)